# Landscape-scale analysis of raccoon rabies surveillance reveals different drivers of disease dynamics across latitude

Matthew Michalska-Smith[1,2‡], Meggan E. Craft[1‡*], Amy J. Davis[3], Amy T. Gilbert[3], Richard B. Chipman[4], Jordona Kirby[4], Kathleen M. Nelson[4], Xiaoyue Ma[5], Ryan Wallace[6], Grace Miller[7], Kim M. Pepin[3]

**1** Department of Ecology, Evolution and Behavior, University of Minnesota, Saint Paul, Minnesota, United States of America, **2** Department of Plant Pathology, University of Minnesota, Saint Paul, Minnesota, United States of America, **3** United States Department of Agriculture, Animal and Plant Health Inspection Service, Wildlife Services, National Wildlife Research Center, Fort Collins, Colorado, United States of America, **4** United States Department of Agriculture, Animal and Plant Health Inspection Service, Wildlife Services, National Rabies Management Program, Concord, New Hampshire, United States of America, **5** Division of Core Laboratory Services and Response, Office of Laboratory Systems and Response, Centers for Disease Control and Prevention, Atlanta, Georgia, United States of America, **6** Poxvirus and Rabies Branch, Division of High-Consequence Pathogens and Pathology, National Center for Emerging and Zoonotic Infectious Disease, Centers for Disease Control and Prevention, Atlanta, Georgia, United States of America, **7** College of Veterinary Medicine, University of Minnesota, Saint Paul, Minnesota United States of America

‡ These authors are joint first-authorship on this work.
* craft004@umn.edu

## Abstract

When raccoon rabies first invaded the mid-Atlantic United States, epizootics were larger, longer, and more pronounced than those in its historic, more southern, range, suggesting a North-South gradient in disease dynamics. In addition, due to higher raccoon densities and concentrated feeding sources, urban areas might sustain larger epizootics, suggesting an urban-rural gradient might likewise influence dynamics. Here we leverage long-term surveillance data on raccoon rabies, collated by the Centers for Disease Control and Prevention, United States Department of Agriculture, and state and local public health agencies to better understand the role of latitude and urbanness for raccoon rabies epizootiology. Our analysis utilizes surveillance data from the 20 states composing the raccoon rabies enzootic area across 2006–2018. We identified effects of latitude and human population density (a proxy for urbanness) on the county-level probability of detecting raccoon rabies. We find that: 1) in the northeastern US, more samples are submitted in the summer, and more positive results are obtained, albeit with a lower likelihood of a given sample being found to be rabid, while these trends are independent of season at southern latitudes; 2) the association between urbanness and risk of rabies cases varies across latitude, with greater rabies presence in rural vs. urban counties in the south and a more consistent risk across urbanness in the north; and 3) the most consistent predictors

purpose. The work is made available under the Creative Commons CC0 public domain dedication.

**Data availability statement:** USA Federal government policy for protecting personally identifiable information regarding disease incidence at the county-month level prohibits sharing of the data set analyzed in this work. Data for this study were obtained from the National Rabies Surveillance System, maintained by the US Centers for Disease Control and Prevention (rabies@cdc.gov) and the Enhanced Rabies Surveillance database of the National Rabies Management program of USDA-APHIS Wildlife Services (aphis.customersupport@usda.gov). Code available online at https://github.com/mjsmith037/Raccoon_Rabies_Across_Urbanness_And_Latitude.

**Funding:** This work was supported by the National Science Foundation [grant number DEB-2321358 to MEC]. This work was supported by the United States Department of Agriculture, Animal and Plant Health Inspection Service, Wildlife Services. The funders had no role in study design, data collection and analysis, decision to publish, or preparation of the manuscript. The findings and conclusions in this manuscript are those of the authors and should not be construed to represent any official USDA or United States Government determination or policy. Any use of trade, firm, or product names is for descriptive purposes only and does not imply endorsement by the United States Government.

**Competing interests:** The authors have declared that no competing interests exist.

of raccoon rabies detection are spatiotemporal effects, suggesting that recent detection of cases in a county or its neighbors are more informative of raccoon rabies dynamics than are general metrics like latitude and urbanness. Statistical and spatial long-term studies like these not only can improve understanding of wildlife disease patterns but can help guide public health and wildlife management efforts in areas most at risk for raccoon rabies virus infection.

## Author summary

Factors such as latitude and level of urbanization can impact pathogen spread in important ways. Here we leverage long-term data on raccoon rabies surveillance in raccoons to better understand the role of latitude and urbanness in the number and distribution of raccoon rabies cases. We find that: 1) in the northeastern US, more surveillance samples for raccoon rabies are collected during the summer months compared to winter months, with a lower proportion of rabid animals during the summer months, in contrast to more consistent trends in southern counties; 2) rabies cases in a given county and year were heavily influenced by previous months' case data from the focal county or its neighboring counties, with additional effects of urbanness, especially at southern latitudes; and 3) raccoon rabies cases are largely independent of the proportion of habitat that is favorable for raccoons, but cases do drop when the proportion of favorable raccoon habitat is at very low levels. Identification of trends in where and when cases are most likely to occur can help with a better understanding of the factors driving endemic disease dynamics in wildlife, and can help focus education, surveillance, and management efforts on higher risk areas and seasons.

## Introduction

Rabies is a highly fatal zoonotic disease with a near global distribution, spanning latitudes and animal host species, including wild carnivores. Despite the existence of post-exposure prophylaxis for humans [1] and vaccines for domestic dogs and wildlife, rabies still causes approximately 60,000 human deaths per year worldwide [2]. An estimated 50,000 human exposures annually in the United States (US) are linked to wildlife reservoir species such as raccoons (*Procyon lotor*) [3–5].

Historically, the variant of *Lyssavirus rabies* (RABV) adapted to raccoons in the United States was isolated to the southeastern portion of the country (i.e., Florida and Georgia) [6]. However, translocation of raccoons to West Virginia during the 1970s unknowingly introduced infected animals to a susceptible raccoon population, leading to epizootic spread of the disease across the eastern United States [7]. Intense management efforts, including the use of oral rabies vaccine (ORV) for raccoons, have been successful in containing the westward and northward spread of raccoon rabies virus [8,9].

Though raccoon RABV now spans the entire—more than 20°—latitudinal range of the eastern United States, including being found across a three order-of-magnitude span of county-level human population densities, the distribution of confirmed cases is far from uniform [10,11]. Some ecological contexts across a landscape are more prone to hotspots of disease emergence/spillover [12], while others facilitate sustained transmission within a wildlife host reservoir [13]. Disease dynamics are also heterogeneous in time; some pathogens are maintained at a constant prevalence (enzootic or endemic dynamics), while others produce sporadic outbreaks (epizootics or epidemic dynamics, where outbreaks of infection are followed by inter-epidemic troughs in case numbers). Epidemic dynamics can be governed by seasonal changes to host population demography or contact rates [14,15] that may differ across latitudinal climes. Raccoon rabies dynamics can be characterized in many ways. One method that reduces biases inherent to the passive surveillance utilized in the case of rabies in the United States is a quantification of "presence," i.e., whether or not there is at least one infected animal within a prescribed area. By incorporating a better understanding of the role of environmental factors such as seasonality and urbanness in raccoon rabies presence at a landscape scale, limited resources can be better prioritized to support enhanced surveillance efforts and refine management strategies across different ecoclimes.

Latitude and seasonality are linked. In lower latitudes, rainfall can vary seasonally (e.g., rainy vs. dry seasons), whereas in higher latitudes, there can be extreme variation in temperature (e.g., summer vs. winter). Seasonal effects can drive animal demography (e.g., births, deaths) and behavior (e.g., mating, birthing, denning, feeding, migration), which can in turn impact disease dynamics via changes in aggregations, contacts, and interactions with infected individuals [14,16]. Accordingly, the number of raccoon rabies cases shows variation in seasonal trends with peaks in rabies occurrence ranging from: March (Florida) [6]; "spring" (March through June) with more sporadic peaks in early fall (October and November: Mid-Atlantic states) [17,18]; late winter (February and March: Virginia) [19]; spring and fall (11 states in the Eastern U.S.) [20]; or just in the fall-winter (Pennsylvania, Ohio, and West Virginia) [21]. Theoretical models show that a combination of wave-like spread of infection and pulsed births in northern climes could create asynchronous dynamics [22]. Additionally, in areas where individuals congregate for warmth in the winter months, pathogen transmission might increase with the change in social contacts and increase in contact duration [14,16,23]. Alternatively, in southern climes where raccoons congregate less and where they breed throughout the year, there could be more consistent transmission throughout the year.

Urbanization can result in the proliferation of urban-adapted species such as raccoons [24], potentially increasing risk of zoonotic infection [25]. For example, in urban areas there are extremely high densities of raccoons [26–31], which can lead to high mixing and contact rates, both between raccoons and with humans [32,33]. This population and contact structure has consequences for transmission: areas with a higher human population density tend to have larger and longer epizootics [34]. Yet, these high-density urban areas may also be fragmented by highways and rivers, which could potentially lead to reduced contact and transmission across the landscape, in contrast to rural areas, where raccoon densities are lower, but potentially less fragmented [35].

Despite this work on the role of latitude and urbanness, we lack a comprehensive framework for understanding how these factors interact, limiting our ability to anticipate the distribution of raccoon rabies across the landscape where it is enzootic. Here we leverage extensive long-term datasets on raccoon rabies surveillance collected as part of enhanced rabies surveillance by the United States Department of Agriculture (USDA) and public health surveillance data collected by states and reported annually to the Centers for Disease Control and Prevention (CDC) to better understand the role of latitude, seasonality, and urbanness in raccoon rabies transmission ecology. The combined data consist of mostly passive surveillance across the entire range of raccoon RABV in the eastern US during 2006–2018. Our study objectives are to quantify the relationship between latitude, seasonality, and "urbanness" (as measured by human population density) on raccoon rabies presence, quantified as the probability of at least one detected case of rabies in raccoons of raccoon rabies in a given county in a given month. We predict that this probability may be higher in northern latitudes (as seasonality can increase viral spread through co-denning) and in urban areas (due to congregations around feeding locations

and high densities of raccoons). We predict that the number of rabies cases in northern ecoregions will cluster more in time (i.e., seasonal winter peaks) than in southern ecoregions, and that seasonal peaks might be more pronounced in the high-density urban areas. Identification of spatial and temporal trends in where cases are most likely to occur can help with a better understanding of the factors driving endemic zoonotic disease dynamics, and can help focus education, surveillance, and management efforts on higher-risk time periods and areas.

## Methods

### Surveillance data and covariates

We used two datasets from the eastern United States which consisted of individually sampled animals that were tested for RABV by Direct Fluorescent Antibody Test (DFA), Direct, Rapid Immunohistochemical Test (dRIT), or both [36,37]. Both of these testing methods are approved and equivalent diagnostic methods [38,39]. Cases of discordance between these tests are rare and usually the result of an indeterminate dRIT result. In these cases, samples are subjected to secondary/confirmatory testing (by DFA or real-time PCR). Any test results that remained inconclusive (0.16% of case reports) were dropped from the analysis. The first dataset is Enhanced Rabies Surveillance (ERS) from USDA's National Rabies Management Program (NRMP) from the years 2006–2018; all of the positives were variant typed to confirm infection with raccoon rabies variant [e.g., 40]. Data included: date and county of collection, species, and result of RABV test (positive or negative). Data from this source combined several methods of enhanced sampling (e.g., strange acting, found dead, roadkill, nuisance) [21]. The second data source was the CDC National Rabies Surveillance System from the years 2006–2018 which comprises passive public health surveillance provided by a network of over 130 state health, agriculture and university laboratories [e.g., 10]. Public health surveillance case data in this region are not routinely variant typed, however terrestrial cases occurring in the eastern U.S. are almost exclusively the raccoon RABV variant [41,42]. The dataset included all variables measured for the USDA data and, additionally, the latitude and longitude of the sample location, which in some states was often the centroid of a county. For this study, we were only interested in raccoons and the raccoon variant of RABV, so we only used reports from raccoons in both datasets. As both datasets fundamentally relied on passive reporting from the public, there was no formal responsiveness of sampling intensity to previously identified positive rabies cases.

To focus on understanding raccoon rabies disease dynamics in the absence of active management, we focused our analysis on county-year combinations that had not been managed with the oral rabies vaccine (ORV) that year, and additionally removed two counties that had exceptionally intensive surveillance (more than 200 tests in our dataset), yet no detected positive cases (Mobile and Washington Counties in AL). These counties were located near the western edge of the raccoon rabies endemic range and near the southern extent of the ORV management zone. As such, we suspected these counties of having outlying management strategies and excluded them (and actively managed counties) in order to focus on areas without management influence [9]. Nevertheless, it is possible that we could still see indirect effects of ORV management policies in unmanaged counties in close physical proximity to managed regions. Because we lacked precise latitude and longitude for all samples, we compiled monthly cases of raccoon RABV at the county level and calculated covariates for each county. The latitude covariate indicated the latitude of the centroid of the county in which the sample was found during surveillance. The seasonality covariate was the month during which the sample was collected. For 'urbanness', we tested various county-level proxies, including: the level of anthropogenic light per county [43]; the level of medium and high development per county (according to National Land Cover Data) [44]; USDA Economic Research Service 2010 Rural-Urban Commuting Area (RUCA) Codes (revised in 2019) [45]; and human population density (calculated by dividing the 2010 US Census population of a county by the land area of that county; hereafter 'population density'). All urbanness metrics produced similar results in exploratory analyses to predict positive raccoon rabies cases. Final models presented here use human population density to represent 'urbanness', as this metric resulted in the lowest Akaike Information Criterion (AIC) within each model type (see following section for details of model formulation).

Another covariate was the proportion of the county containing raccoon favorable habitat (i.e., National Land Cover Database percent land cover of deciduous or mixed forest using Natural Resource Management Plan land cover classes) [21,30,44], which we used as a county-level proxy of raccoon density.

Because the number of cases in a county could be related to the number of cases in previous months, we calculated a temporal lag covariate for each county-month by averaging the number of raccoon RABV cases during the previous three months per focal county. Although RABV infectious periods in raccoons are approximately 1 week [46], the incubation period can be up to 19 weeks [47]; therefore, a time period of three months could represent an opportunity for onward transmission within a county [21]. To account for spatial effects of cases in neighboring counties (where high case reports in surrounding counties in the past might predict the number of current cases), we calculated a spatial effect covariate for each county-month by averaging the number of cases found in the current and previous two months in surrounding counties (i.e., those sharing a border with the focal county).

## Model formulation

We constructed generalized linear mixed models (GLMM) to quantify the effects of covariates on three different raccoon RABV outcomes. In the main text, we focus on a response variable of county-level probability of at least one raccoon RABV case report in a given county per month (i.e., apparent presence), which was modeled using a Bernoulli-distributed (1 if raccoon rabies was present in a county in a given month, 0 if not) GLMM. In S1 Text, we additionally present results for monthly proportion positive (the proportion of total samples that are positive for raccoon RABV in a given county-month), which was modeled using a beta-distributed GLMM, and apparent persistence (whether raccoon RABV was continually and repeatedly detected in a county through time or not), which was modeled using a Bernoulli-distributed (1 if raccoon rabies was consistently present in a county, 0 if not) GLMM. We use the statistically appropriate term 'apparent' throughout this paragraph to highlight that these response variables are biased samples of the 'true' presence or persistence because our analysis does not account for error in the observation process. From here onwards, we will refer to presence, proportion positive, and persistence for conciseness.

For all models, we accounted for surveillance effort across counties and months using a fixed effect of the log-transformed total number of tests per county per month plus one. We additionally allow for potential systematic differences in surveillance across counties by including county identity as a random effect. We considered explicit splining functions for the density, latitude, and month covariates to account for nonlinearities in effects across the dataset. After initial exploratory analysis, we included interactions between latitude and density and latitude and month (to account for different seasonal effects across latitudes). We identified the combination of variables that best explained each of presence, proportion positive, and persistence, including the specific form of each splining function, using AIC model selection. When plotting model predictions, we generate 1000 synthetic data points by drawing values (with replacement) from each variable independently and replicating these data points across a range of values for parameters of interest (i.e., month, latitude, and density).

These models were fit in R (version 4.1.0) [48], making use of the GLMMadaptive [49], rstatix [50], tidyverse [51], ggmap [52], ggbeeswarm [53], ggpubr [54], patchwork [55], kableExtra [56], XML [57], and tigris [58] packages. Code available online at https://github.com/mjsmith037/Raccoon_Rabies_Across_Urbanness_And_Latitude.

## Habitat piecewise linear models

Because we observed a discrete transition in the slope of the relationship between the proportion favorable habitat and our response variables, we fit a piecewise linear model to the relationship, reporting the spearman correlation coefficient and linear regression-associated p-value for each of two components (above and below a threshold level of raccoon-favorable habitat coverage). To identify an informative breakpoint, we considered all breakpoints between 0 and 1, at increments of 0.01, and chose the breakpoint for each response variable (presence, proportion positive, and persistence)

that corresponded to the maximal correlation coefficient for the first (positively correlated) portion of the data, i.e., at low levels of raccoon-favorable habitat coverage. This resulted in breakpoints of 0.16, 0.17, and 0.16 for presence, proportion positive, and persistence, respectively.

## Results

### Rabies sample submissions and observed proportion positive varies seasonally and across human population densities

Rabies case data represents surveillance from 634 counties and 20 states (Alabama, Connecticut, Delaware, Florida, Georgia, Louisiana, Massachusetts, Maryland, Maine, Mississippi, North Carolina, New Hampshire, New Jersey, New York, Pennsylvania, Rhode Island, South Carolina, Virginia, Vermont, and West Virginia) along the eastern United States over a 13-year period and included a total of ~43,000 samples with ~18,000 rabid raccoons (Fig 1, left). Human population density also varies by county and latitude (Fig 1, center).

In the south (defined as the lowest quartile of counties by latitude), the number of samples submitted and the proportion positive were relatively constant through time (Fig 2, first column). In contrast, more samples were submitted in the north (highest quartile of counties by latitude), both in general (Fig A in S1 Text), and particularly during the summer months (i.e., May-July) than in the winter months (i.e., November-January; Fig 2, top right panel). Proportion positive also varied seasonally with lower raccoon RABV proportion positive in the north in the summer months than in the winter months (Fig 2, bottom right panel). Thus, in the north more surveillance samples for raccoon rabies were collected, yet with a lower proportion of rabid animals during the summer months compared to winter months. Total samples submissions

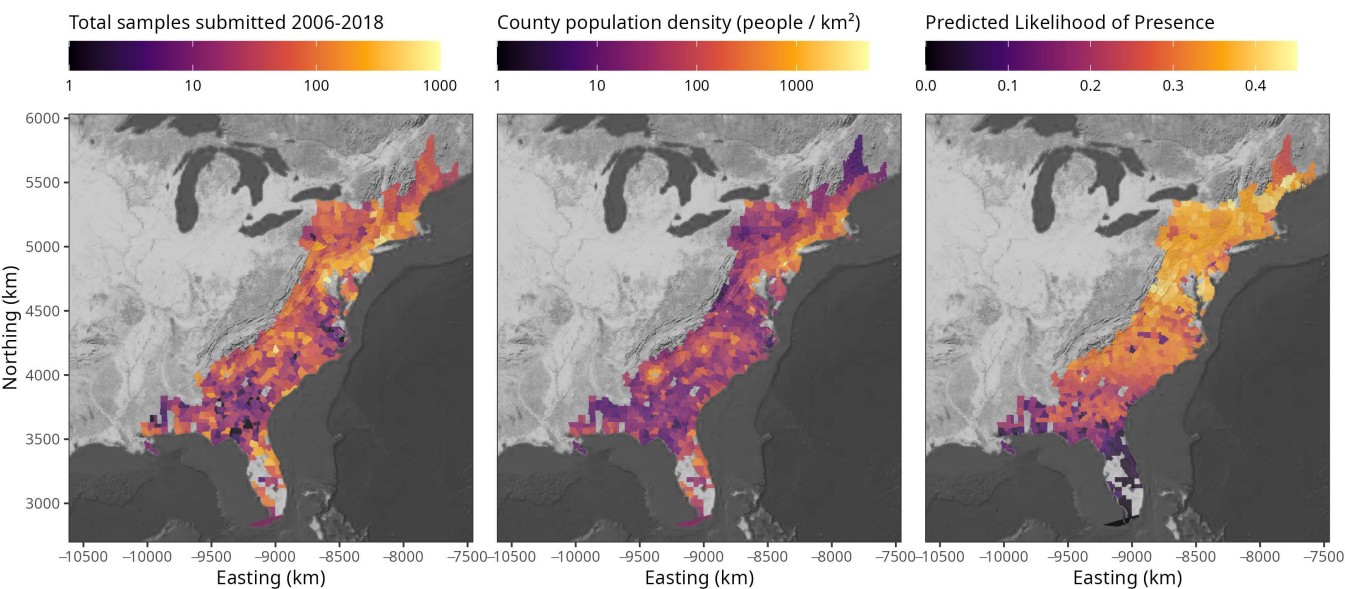

**Fig 1. County-level maps of raccoon rabies surveillance data, human density, and model predicted likelihood of raccoon rabies presence.** Raccoon rabies surveillance varied spatially, with more samples submitted from urban areas and in the north (left). County population density of humans (middle). Average monthly predicted presence of at least one raccoon rabies case per month, where detection is influenced by interactions between latitude and month, as well as latitude and population density; there were also significant effects of temporal lag, spatial lag, and total number of tests conducted (right; Table 1). Note the color scales differ between panels. Analogous figures to the right panel are available for rabies proportion positive and persistence in Fig D in S1 Text. Basemaps are U.S. Geological Survey topographic maps, generated with the basemaps package [59] in R (version 4.1.0) [48].

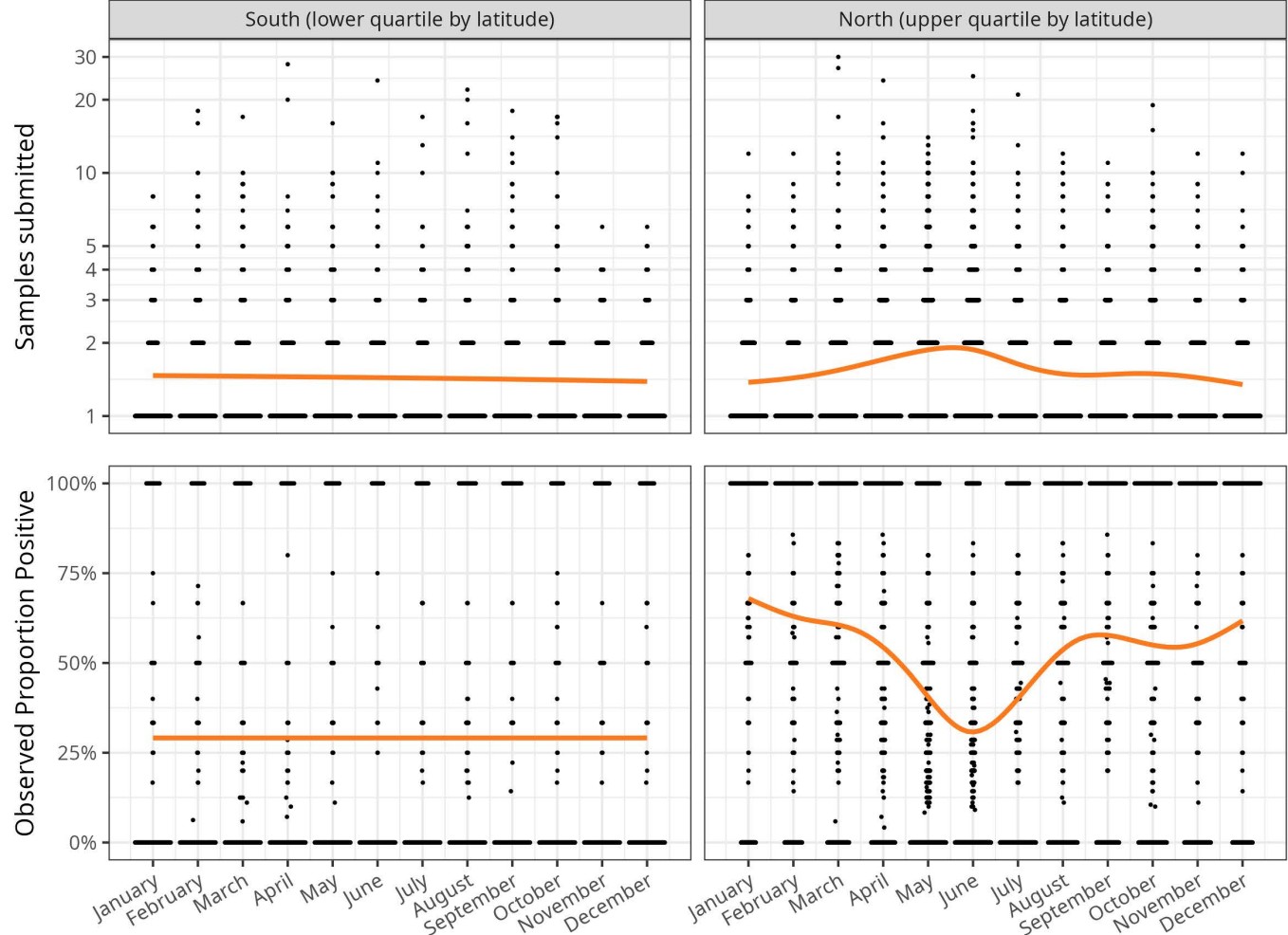

**Fig 2. The number and observed proportion positive of raccoon rabies surveillance samples varied seasonally in the north but not in the south.** Row 1 illustrates the number of tests conducted per county per month across all years for the most southern and most northern counties in the dataset. Row 2 illustrates the proportion of tests found to be positive per county per month across all years for the same counties. The number of tests and observed proportion positive from the mid-quartile latitude ranges are intermediate to the extremes presented here and are not presented. The width of the black points/bars represents the number of county-months with a given number of tests or proportion positive, while the orange line is a best-fitting generalized additive model.

also increased with human population density, with the slope of the relationship and explanatory power increasing with decreasing latitude (Fig B in S1 Text).

## Latitude, seasonality, and urbanness influence rabies presence

The model with the lowest AIC for predicting the county-level probability of a rabies case report included significant interactions between latitude and month (seasonality), as well as latitude and human population density (urbanness). There were also significant primary effects of latitude and human population density, as well as effects of temporal lag, spatial effect, and total number of tests (Table 1). In general, there was a trend that northern counties detected rabies more than southern counties (Figs 1, right, and 3) and middle-high latitude counties have higher rates of rabies detection than southern counties at moderate population densities (Figs 1, right, and 4).

**Table 1. Best fit models for apparent raccoon rabies presence per county-month.**

| Term | Significance |
| --- | --- |
| (Intercept) | * |
| Month | |
| Latitude | *** |
| Population Density | *** |
| Raccoon Favorable Habitat | |
| Temporal Lag | *** |
| Spatial Effect | *** |
| log(Total Number of Tests Conducted + 1) | *** |
| Latitude:Population Density | *** |
| Month:Latitude | * |

In cases where a spline is fit in the model (i.e., Month, Latitude, Population Density, and, consequently, their interaction terms (indicated with ":")), multiple estimates and p-values are associated with each term. Here, we only report the significance of the smallest associated p-value as an indication of the overall significance of that term, with *, **, and *** indicating p-values less than 0.05, 0.01, and 0.001, respectively. The details of each fitted spline for rabies presence can be found in Table A in S1 Text. An analogous summary for all three response variables can be found in Table B in S1 Text, with fitted spline details for persistence and proportion positive in Tables C and D in S1 Text, respectively.

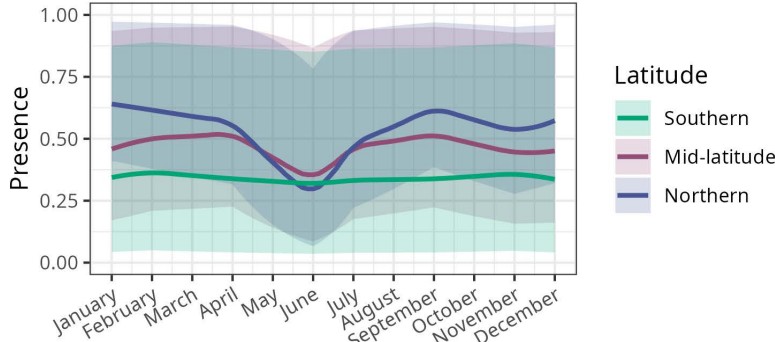

**Fig 3. Significant interaction between latitude and month on the predicted probability of raccoon rabies being detected (presence).** The horizontal axis represents the month of the year, while the vertical axis is the model predicted probability of having at least one positive sample in a given county. Three example latitudes spanning the south (30°N), mid-latitude (35°N), and north (40°N) are displayed. Confidence intervals depict the 10th to 90th quantile of model predictions for 1000 bootstrapped data points for each combination of latitude and month (see methods). Analogous figures for rabies proportion positive and persistence can be found in Fig E in S1 Text.

However, due to the significant interaction between latitude and month on the county-level monthly probability of a raccoon rabies case report, the presence of a case report was not always higher in the north than in the south, especially during the summer months. When we plotted the monthly predicted presence for a range of latitudes, the detection probability increased with latitude in most cases, and northern climes had more seasonal variation (Fig 3).

The effect of latitude was also modulated through significant interactions with human population density. Southern counties tended to have lower probabilities of detecting raccoon rabies, especially at intermediate population densities of 100–300 people/ km², while central and northern counties were more similar (Fig 4). At high population densities, however, we see southern and northern counties converging to have higher raccoon rabies detection probabilities than more central counties. Note, however, there were very few counties with such high population densities.

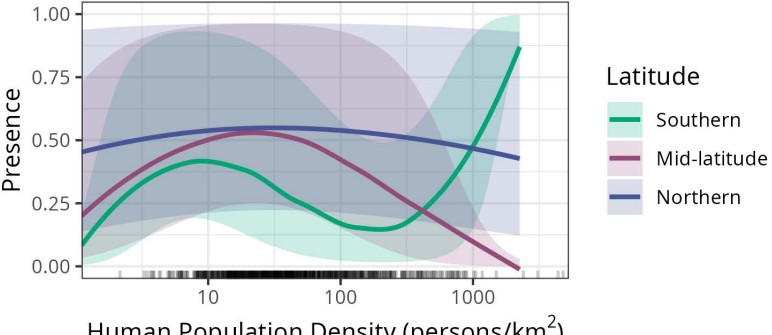

**Fig 4. Significant interaction between latitude and population density on the predicted probability of raccoon rabies being detected (presence).** Human population density is plotted on the horizontal axis, including a rug to indicate where the counties used in this analysis fall in the range of densities. The vertical axis is the model predicted probability of having at least one positive sample in a given county. Three example latitudes spanning the south (30°N), mid-latitude (35°N), and north (40°N) are displayed. Confidence intervals depict the 10th to 90th quantile of model predictions for 1000 bootstrapped data points for each combination of latitude and density (see methods). Analogous figures for rabies proportion positive and persistence can be found in Fig F in S1 Text.

## Raccoon favorable habitat has a threshold effect on rabies cases

Although the relationships between raccoon favorable habitat and raccoon rabies dynamics were not statistically significant when including their whole range of values, a closer look at these data reveal a potential threshold effect. When there is relatively low (<16%) coverage of raccoon favorable habitat (proportion deciduous or mixed forest land cover), there is a nearly 25-times greater relationship with rabies presence (and a corresponding nearly 50-times increase in explained variance), compared to favorable habitat coverage of greater than 16%. That is, for most levels of habitat there is no relationship between raccoon rabies presence and favorable habitat coverage, but if a county has very little available favorable habitat, we see a reduction in the likelihood of detecting raccoon rabies in a given month (Fig 5). In light of this result, the

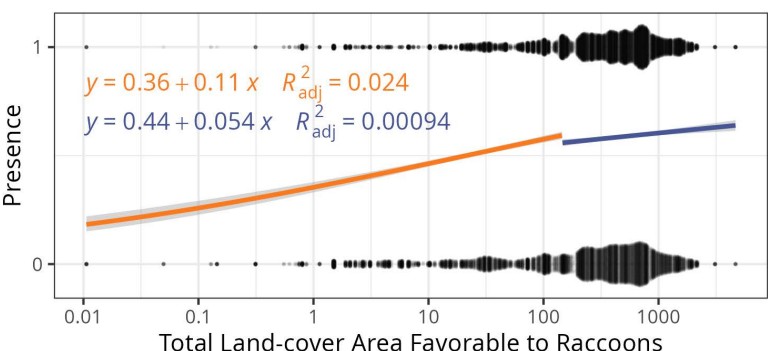

**Fig 5. Favorable raccoon habitat is positively related to raccoon rabies presence at very low levels of favorable habitat per county.** The raw data for the amount of favorable raccoon habitat by presence (whether or not a county has at least one positive sample in a given month) is indicated with black points, which are jittered vertically around their binary values to indicate the frequency of counties with those habitat values. Two predicted lines are drawn, an orange one which illustrates the positive relationship between habitat and outcome at low levels of raccoon-favorable habitat (p-value < 0.001), and a blue one illustrating that above a minimum threshold of available raccoon habitat, rabies is nearly equally likely to be detected (p-value = 0.152). While the overall variance explained is low for both low and high levels of habitat coverage, the R2 value for habitat coverage less than 16% is nearly 50 times that for habitat values greater than 16%. We find similar results for rabies proportion positive and persistence (Fig G in S1 Text).

insignificant relationship across the whole range of favorable habitat coverage is not surprising, as piecewise relationships are not typically able to be captured by standard GLMM approaches, as used in this work.

## Discussion

Here we use surveillance data to evaluate whether latitude and urbanness impact the probability of a raccoon rabies case report ("presence"). We find that county-level raccoon rabies dynamics are strongly driven by temporal (the number of rabies cases during the previous three months in that county) and spatial (caseloads in surrounding counties in the current and previous two months) context, as well as testing intensity (total number of tests conducted in a given month), supporting the development of criteria for sample prioritization such as the epizootiological importance [38,42,60]. Additionally, however, we also find evidence for the role of latitude and urbanness (as measured by human population density), especially through interactions between latitude and season ("month") and between latitude and population density. These results suggest that there is an additional benefit to enhancing raccoon rabies surveillance, management, and education efforts in those particular counties and/or months for which additional surveillance is thought to be most informative. Areas of predicted high risk would additionally be candidates for follow-up epidemiological investigations.

In the absence of a county-level density map of raccoons, we used the proportion of a county's land cover consisting of habitat that has previously been associated with raccoon preference [61–63] and, consequently, RABV prevalence [21] as a county-level proxy of locations we are likely to see infected raccoons. In agreement with prior literature [21], we found that in counties with less than 16% land cover of deciduous or mixed forest, there were fewer raccoon rabies cases. Thus, while we did not see a trend across the whole range of favorable habitat coverage, wildlife managers could view counties with 'unfavorable' habitat (meaning less than 16% of land-cover being favorable to raccoons) as being most amenable to limiting rabies spread in the absence of ORV control. Likewise, in the face of land-use change, changes to raccoon favorable habitat coverage are unlikely to have an effect unless it is already quite limited. The habitat relationships explored here, however, should be interpreted with some caution as our binary classification of the suitability of land cover is an oversimplification both of raccoon space use [30,63,64] and of the importance of raccoon density on RABV dynamics [30].

We predicted that rabies cases in northern ecoregions would be more clustered in time than in southern ecoregions; indeed, in the north we found seasonal trends with more raccoon rabies cases detected in the winter than in the summer. However, in the north, more raccoons were submitted for rabies surveillance in the summer compared to the winter; therefore, inferring case trends from looking only at the number of positive samples through time could be misleading. In the north, more raccoons might be submitted for testing in the summer than in the winter because humans are more likely to be outside in the summer (a consequence of passive surveillance that relies on public reporting) [65] when raccoons are more likely to be active with their weaned young [66]. Our results suggest that rabies might be circulating more in the winter in the north (consistent with previous studies) [21], albeit with the caveat that the month of rabies detection does not necessarily match the month in which the animal was infected. While the National Rabies Surveillance System (NRSS) is responsive to human exposure risks as outlined by the CSTE [38,42,60], there may be a reduced need for enhanced raccoon rabies surveillance (e.g., by USDA) during the summer months in the north [8].

We predicted that the probability of a raccoon rabies case report would be higher in the northern latitudes. We found this to be broadly true, as there were significant interactions between latitude and month leading to higher raccoon rabies presence as latitude increases in the fall, winter, and spring. The surprise here was the notable drop in raccoon rabies presence in the summer in the far north, despite increased sampling. This could be due to synchronous cases occurring during the winter months, leading to a reduction in potentially infectious interactions in the summer, as might be expected for the dynamics of an acute disease with almost-always fatal outcomes in a spatially structured population [67]. There were also significant interactions between latitude and population density (urbanness), with generally higher raccoon rabies presence in northern climes than southern climes for suburban and urban counties. This nuance could guide the distribution of educational resources in the north (more urban/suburban) vs. the south (more rural) and highlights the

importance of adaptive surveillance [68–70]. In our analysis, we aggregated across years, possibly obscuring an effect of the time-since-introduction, which varies across the spatial extent of our data. In particular, one might expect that the relatively recent introduction of rabies to the northern states (i.e., within the past 50 years) might be consequential to disease dynamics. We do not see any evidence for this in our data. While the overall number of submitted samples and positive cases shows a slight decrease over the thirteen years of our data, the slope of this trend is independent of latitude (Fig C in S1 Text).

Urban areas are important for public health and wildlife managers because not only are there more opportunities for rabid raccoons to interact with high densities of humans and associated companion animals, but urban areas are more challenging and expensive to manage with oral vaccine baiting [71,72]. Although we predicted that the probability of a raccoon rabies case report would be higher in urban areas, our results are complicated because of the interaction with latitude and important sampling limitations. In general, there was slightly higher raccoon rabies presence in counties with intermediate human population density in the mid-latitude and northern latitudes; this could potentially be explained by more continuous swathes of favorable raccoon habitat in counties with intermediate population density than in high population density, but still many humans to report raccoons. What was contrary to our expectations was the higher raccoon rabies presence in rural (low human population density) counties, compared to urban counties (with the exception of very high human density counties in the south). We expected that because urban areas have high densities of raccoons, there are more opportunities for human-raccoon interactions, leading to more opportunities for a sample to be tested (and hence for rabies to be detected) in higher density settings. One potentially contributing factor is spatial and demographic bias in suspected rabies sample submission [73,74]. For instance, people in rural areas might be more aware of when an animal is acting strangely, increasing the likelihood that a submitted sample would be found rabid, should it rise to the standard of warranting submission. In contrast, submissions from urban areas might be more independent of the likelihood of infection. Increased and targeted education could reduce this discrepancy, by levelling *a priori* expectations between urban and rural populations.

Our counterintuitive urbanness results might have to do with challenging sampling issues. The surveillance data were mostly passive and states and counties may vary in surveillance intensity and consistency [11,75]. For example, raccoon rabies is more recently endemic in the north, perhaps with consequently more recent education, awareness, and reporting; this could have led to the northern counties being more uniformly sampled than southern counties. Passive surveillance usually underestimates true disease prevalence [76], and can lead to reporting issues which vary by raccoon and human density. For instance, raccoons are often found in high densities close to human areas (i.e., urban areas), which could potentially account for higher rates of submission for rabies testing in urban areas [41]. Conversely, it is hard to tell if raccoon rabies is not present in a county with low human population density, or if surveillance may be inadequate [11]. We indeed see this trend in our county-level data – counties with more people had more raccoons submitted for testing, while there were fewer submissions from rural counties, especially in the south (Fig C in S1 Text).

In addition, counties are likely not the best spatial resolution for the study of disease dynamics [11]. Aggregating samples at the county-level may be too broad for estimation of some covariate relationships. For example, counties vary in size and shape (especially in the south), and information can be lost by averaging widely varied habitat types and human population densities over the county scale. Though, it is important to note that we did not see any explanatory relationships between county land area and rabies presence. Likewise, the borders of counties are delineated according to a combination of terrain, anthropogenic features, and political boundaries. While some of these might have consequences for animal movement and/or disease spread, others are more permeable. There could be some misclassification bias as the county of detection might not be the county of infection and the month of detection might be not scale precisely with the month of infection due to the variable rabies incubation period, however we expect similar misclassification across space and time which would add additional noise to our results. Future work looking into how these results might map onto a more granular view of the spread of disease is needed [11].

The distribution of "urbanness" is also heterogeneous within counties, and county boundaries and urban areas do not necessarily align. We tried to mitigate this challenge by using model selection to choose from a variety of plausible urbanness proxies, but because there is no robust metric for urbanness at the sub-county level, we should view our urbanness results with caution. Having access to exact sample locations, or at least townships, would be ideal. In 2022, the Council of State and Territorial Epidemiologists voted to add sub-county location data to the list of recommended variables for rabies reporting (to the state) and notifications (to CDC) [38]. Spatial understanding of rabies cases analyzed through passive surveillance data systems, and consequent potential for spatial targeting of management and interventions, may therefore improve in future years.

Despite the challenges studying urbanness with county-level data, ultimately, we did find a significant interaction between human population density and latitude on rabies presence, in addition to the more expected relationships with temporal lag, spatial effect, and total number of tests. Rabies is more likely to be detected in counties that had positive tests in previous months and/or in neighboring counties in the recent past, in counties with greater sampling effort. Yet, rabies dynamics also have complex relationships with latitude, urbanness, and seasonality, impeding generation of generalizable predictions across large-scale landscapes. These are complex systems with trends that are less absolute and more nuanced. Identification of temporal and spatial trends in where cases are most likely to occur can help with a better understanding of the factors driving endemic zoonotic disease dynamics, and can help focus public education, surveillance, and management efforts on higher-risk time periods and areas [77].

## Supporting information

**S1 Text. Supporting information including figures and tables to supplement the claims in the main text, analogous figures and tables for alternative response variables to those used in the main text, and additional methods and discussion associated with these alternative response variables.**
(PDF)

## Author contributions

**Conceptualization:** Matthew Michalska-Smith, Meggan E. Craft, Amy J. Davis, Amy T. Gilbert, Kim M. Pepin.

**Data curation:** Matthew Michalska-Smith.

**Formal analysis:** Matthew Michalska-Smith, Amy J. Davis.

**Funding acquisition:** Meggan E. Craft.

**Investigation:** Meggan E. Craft.

**Methodology:** Matthew Michalska-Smith, Meggan E. Craft, Amy J. Davis, Amy T. Gilbert, Kim M. Pepin.

**Project administration:** Meggan E. Craft.

**Supervision:** Meggan E. Craft, Amy J. Davis, Amy T. Gilbert, Kim M. Pepin.

**Visualization:** Matthew Michalska-Smith.

**Writing – original draft:** Matthew Michalska-Smith, Meggan E. Craft, Grace Miller.

**Writing – review & editing:** Matthew Michalska-Smith, Meggan E. Craft, Amy J. Davis, Amy T. Gilbert, Richard B. Chipman, Jordona Kirby, Kathleen M. Nelson, Xiaoyue Ma, Ryan Wallace, Grace Miller, Kim M. Pepin.

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
