## [Decision Letter · Decision Letter 0]

25 Feb 2025

PNTD-D-24-01864

Landscape-scale analysis of raccoon rabies surveillance reveals different drivers of disease dynamics across latitude

Dear Dr. Michalska-Smith,

Thank you for submitting your manuscript to PLOS Neglected Tropical Diseases. based on the feedback from the reviewers, we feel that it has merit but requires some revision. Therefore, we invite you to submit a revised version of the manuscript that addresses the points raised during the review process.

Please submit your revised manuscript within 60 days Apr 26 2025 11:59PM. If you will need more time than this to complete your revisions, please reply to this message or contact the journal office at plosntds@plos.org. Please include the following items when submitting your revised manuscript:

We look forward to receiving your revised manuscript.

Kind regards,

Simon Rayner

Academic Editor

David Safronetz

Section Editor

Shaden Kamhawi

co-Editor-in-Chief

Paul Brindley

co-Editor-in-Chief

**Journal Requirements:**

1) We noticed that you used the phrase 'data not shown' in the manuscript. We do not allow these references, as the PLOS data access policy requires that all data be either published with the manuscript or made available in a publicly accessible database. Please amend the supplementary material to include the referenced data or remove the references.

2) Some material included in your submission may be copyrighted. According to PLOSu2019s copyright policy, authors who use figures or other material (e.g., graphics, clipart, maps) from another author or copyright holder must demonstrate or obtain permission to publish this material under the Creative Commons Attribution 4.0 International (CC BY 4.0) License used by PLOS journals. Please closely review the details of PLOSu2019s copyright requirements here: PLOS Licenses and Copyright. If you need to request permissions from a copyright holder, you may use PLOS's Copyright Content Permission form.

Potential Copyright Issues:

- Figures 1 and S1. Please provide a direct link to the base layer of the map (i.e., the country or region border shape) and ensure this is also included in the figure legend; and provide a link to the terms of use / license information for the base layer image or shapefile. We cannot publish proprietary or copyrighted maps (e.g. Google Maps, Mapquest) and the terms of use for your map base layer must be compatible with our CC BY 4.0 license.

3) We note that your Data Availability Statement is currently as follows: "USA Federal government policy for protecting personally identifiable information regarding disease incidence at the county-month level prohibits sharing of the data set analyzed in this work. Data for this study were obtained from the National Rabies Surveillance System, maintained by the US Centers for Disease Control and Prevention (rabies@cdc.gov) and the Enhanced Rabies Surveillance database of the National Rabies Management program of USDA-APHIS Wildlife Services, (aphis.customersupport@usda.gov). Data can also be obtained through contacting the co-authors.". Please confirm at this time whether or not your submission contains all raw data required to replicate the results of your study. Authors must share the “minimal data set” for their submission. PLOS defines the minimal data set to consist of the data required to replicate all study findings reported in the article, as well as related metadata and methods (https://journals.plos.org/plosone/s/data-availability#loc-minimal-data-set-definition).

- The points extracted from images for analysis..

4) Please ensure that the funders and grant numbers match between the Financial Disclosure field and the Funding Information tab in your submission form. Note that the funders must be provided in the same order in both places as well.

**Reviewers' Comments:**

Reviewer's Responses to Questions

**Key Review Criteria Required for Acceptance?**

**Methods** :

-Are the objectives of the study clearly articulated with a clear testable hypothesis stated?

-Is the study design appropriate to address the stated objectives?

-Is the population clearly described and appropriate for the hypothesis being tested?

-Is the sample size sufficient to ensure adequate power to address the hypothesis being tested?

-Were correct statistical analysis used to support conclusions?

-Are there concerns about ethical or regulatory requirements being met?

Reviewer #1: The paragraph (ll. 51-52) states that surveillance data from the 20 states is used for this analysis. It is recommended to clarify which states are being referred to. At the start of the summary, the authors refer to the Northeastern/mid-Atlantic US; however, I was unable to find a definition that matches the number of states. This may confuse other readers as well. Since it is important to understand the defined geographical area under surveillance, please consider describing this more clearly.

Regarding county-level data: what is the size of a county, and are the boundaries influenced by natural features, such as rivers? It would be interesting to understand if there was a difference in counties where borders were supported by natural structures regarding transmission. This raises questions about the impact of such features on inter-county species/viral spread.

Have the investigators considered assessing the geographical area under surveillance using a grid system along latitudes or categorizing the existing counties or other sector levels? This would allow them to weigh factors such as natural aspects (e.g., biodiversity, including areas rich in wildlife) and host diversity, including host species density (independent of urbanization density). It may also be useful to consider animal migration routes and climate (if applicable), comparing the different seasons that significantly differ across the states (e.g., is there an increase in cases in areas with longer winters versus areas with no or shorter winter periods?).

It is mentioned (ll. 159-160) that "seasonality can increase spread through co-denning." I would suggest adding "viral" to clarify that seasonality increases the viral spread among raccoon populations due to their co-denning behavior.

Reviewer #2: Lines 150-152: The passive surveillance data source is a major issue for results and conclusions

Lines 154-157: “1) Presence: probability of at least one rabies case in raccoons (positive rabies sample), 2) Prevalence: the proportion of raccoon rabies surveillance samples that test positive, and 3) Persistence: the likelihood of persistent raccoon rabies occurrence (defined as at least n months with a positive sample over the last m months).” #2 and #3 are particularly biased measures. Considered whether it’s best to focus on #1 to improve validity and interpretation.

Lines 160-167: It is good to have the null hypotheses stated clearly! However, as per the above comments. These might need to be simplified.

Lines 173-175: Two different tests are included in the surveillance data that are used; is equivalent test accuracy assumed when tests are used in parallel? In other words, is there any differential diagnostic bias?

Line 178: county of collection is not equivalent to county of infection. Please justify that this is not an important source of bias.

Line 179: in the case of two tests with discordant results, is the sample assumed to be positive? Please clarify.

Lines 185-188: latitude and longitude of the sample location, in some states it was often the centroid of a county. There is an inconsistency in spatial accuracy of the data that requires clarification and justification.

Lines 188-189: “we also removed any data with inconclusive rabies diagnostic test results.” An analysis of the randomness of inconclusive test results would be helpful.

Lines 190-195: “we focused our analysis on county-year combinations that had not been managed with the oral rabies vaccine (ORV) that year … removed counties with rare detection, yet high-surveillance, at the edges of the current RABV range as detection probability was likely higher due to management and our intent was to focus on areas without management influence”. These exclusions are problematic. It seems that the motivation is to only focus on “endemic” raccoon rabies areas, but using surveillance data is in itself, a circular argument. The results would have more credibility if some independent classification of endemic raccoon rabes areas was used, even if this means a substantial reduction in the amount of data analysed. As it is, seems like the data are manipulated to fit the analytical approach. There are also definitional questions – what is ‘rare detection’, what is ‘high-surveillance’ and what is ‘current’? Again, it is worthwhile considering the focus of this study in terms of presence vs. persistence vs. incursion and spread. Disease management influence probably exists throughout the dataset. What are the effects of these data exclusions? Would need to see a sensitivity analysis to be convinced this approach is valid and reasonable.

Lines 197-199: Different county size and shapes; centroid does not represent the location adequately. Why not use actual lat/long when available to reduce this bias?

Line 199: The seasonality covariate was the month during which the sample was collected. This assumes a constant (and short) incubation period. Justify this assumption with some biological reasoning.

Line 209: Given the county shape variation, total area of habitat should also have been evaluated in addition to proportion of the county containing raccoon favorable habitat.

Lines 215-216: Given the great variation in RABV infectious periods in raccoons (from 1 week to up to 19 weeks), perhaps some exploration of the temporal patterns in the data using time series analysis is warranted. Also, doesn’t this mean there could be misclassification bias when using month of collection in the analysis?

Lines 218-219: “(where high case reports in surrounding counties in the past might predict the number of current cases)” sounds more like surveillance bias than disease spread, but in reality, is impossible to disentangle. But this is a critical issue – are you analysing disease spread or surveillance effort?

Line 221: “(i.e., those sharing a border with the focal county)”, which to be sounds like a geostatistical rook contingency; a spatial distance buffer might be better, given the arrangement of county shapes.

Line 228: “monthly apparent prevalence”. How did you account for the (likely) great variation in the denominator?

Lines 233-238: definitions of persistence are confusing. My interpretation is that 2 out of 6 months would be defined as persistent, which seems a bit of a liberal definition. Is there any standard definition in this field that can be sued? If not, then some sort of sensitivity analysis should be undertaken, to observe if the definition makes a big difference to the analysis outcome. (Only of course if you plan to retain the persistence analysis in this manuscript).

Lines 238-241: “We use the statistically appropriate term ‘apparent’ throughout this paragraph to highlight that these response variables are biased samples of the ‘true’ presence, prevalence, or persistence because our analysis does not account for error in the observation process.” This is a big problem! Relly, you should only be using the term “proportion positive”. With the surveillance data you have you cannot estimate prevalence, so that use of this term is misleading.

Line 243: including the total number of tests per county per month partly controls for sampling, but at a monthly level testing will fluctuate a lot. But this also makes the findings with respect to the log(Total Number of Tests Conducted + 1) variable difficult to interpret. Or does this sentence mean that sampling was included (forced into) all models as an attempt to adjust for sampling? This needs to be clarified.

**Results** :

-Does the analysis presented match the analysis plan?

-Are the results clearly and completely presented?

-Are the figures (Tables, Images) of sufficient quality for clarity?

Reviewer #1: Surveillance Data:

The surveillance dataset relies on enhanced sampling methods and passive surveillance, which means there is a high likelihood of underestimation. Even though the exclusion of counties with oral rabies vaccine (ORV) management is intended to minimize bias, it could also overlook or miss potentially indirect effects of these management practices on rabies dynamics in nearby areas (counties?) or in seasons without vaccination. A brief explanation of this exclusion reasoning would be beneficial to add, or acknowledging such considerations in the study limitations section.

Reviewer #2: Novelty of results. I feel that this tells us only about the surveillance effort i.e. modelling rabies surveillance, not rabies occurrence. The two are not the same, due to sampling bias.

Lines 292-293: “Thus, in the north more samples are submitted in the summer months when raccoon rabies is less likely to be detected.” This is a peculiar finding that is specific to prevalence, and I suspect it is an artifact of the passive surveillance data that us used in this study.

Lines 351-353: “At high population densities … we see southern and northern counties converging to have higher raccoon rabies detection probabilities than more central counties.” This is an important observation for control and surveillance, and perhaps this study is more suited to making recommendations for surveillance than for understanding rabies epidemiology in this region.

Lines 368-369: “we found a general negative relationship between prevalence and population density”. Again, there are implications for rabies surveillance, rather than insights into rabies epidemiology.

Lines 374-376: “Counties with ‘persistent’ raccoon rabies were defined as having at least 5 months with a rabid raccoon in the previous 11 months and approximately 25% of county-months in our dataset met this criterion.” If persistence remains as an outcome in this study, then it is important to clarify and justify the definition that is used. This is especially important given the acknowledgement that ‘Because we averaged raccoon rabies cases over a broader time period, our outcome of persistence was a coarser measure than outcomes with the presence and prevalence models.’

Lines 383-393: “Although the relationships between raccoon favorable habitat and presence, prevalence, and persistence were not statistically significant in the top models, we did find an interesting threshold effect.” I don’t see value in drawing these conclusions given that the analysis was not statistically significant. This section should be deleted.

**Conclusions** :

-Are the conclusions supported by the data presented?

-Are the limitations of analysis clearly described?

-Do the authors discuss how these data can be helpful to advance our understanding of the topic under study?

-Is public health relevance addressed?

Reviewer #1: The interactions between findings such as latitude/month and rabies presence and prevalence have been well discussed. Particularly, referring to "favorable habitat" in counties without raccoon density data was an interesting angle. Although the introduction indicates that the study results will provide further evidence to inform public health measures, very limited conclusions have been made so far. These valuable insights into disease dynamics could facilitate more informed suggestions for shaping education, surveillance, and management efforts.

Reviewer #2: The study described in this manuscript addresses an important topic that will be of interest to readers of PNTD. The amount of data included in the study (~43,000 samples with ~18,000 rabid raccoons) is considerable, and the analysis methods applied are in general both appropriate and impressive.

The major weakness of the study is the data source: the data used is derived via passive surveillance, conducted at county- and state-level. Such surveillance data is likely subject to enormous bias in terms of the detection of cases and the sampling of detected cases, and as well as possibly the diagnosis of cases. The authors do acknowledge these biases in their discussion. However, the reader is left wondering if the results are valid, given the data source and it’s known bias. I am not convinced that the results of this study provide insights into rabies occurrence, or if it is an analysis of rabies surveillance. As such, the finding that raccoon rabies in the eastern US is related to time, place and sampling effort confirms what is already known. Part of the problem is the inclusion of prevalence and persistence as outcomes of interest in this study. Both of these outcomes are very difficult to measure validly with the data available. Although I appreciate the authors intent for including these outcomes in addition to presence, they become a distraction from the analysis of presence data.

My suggestion is to simplify the analysis by restricting it to rabies presence, and then to more clearly make the argument why the type of data support the outcomes of that analysis.

**Editorial and Data Presentation Modifications?**

Reviewer #1: (No Response)

Reviewer #2: (No Response)

**Summary and General Comments** :

Reviewer #1: (No Response)

Reviewer #2: Abstract

Include some details of quantitative results in the abstract e.g. measures of autocorrelation, measures of association.

Lines 62-64: what are these areas? Provide some more detailed information.

Line 66: latitude is described as an environmental factor. Please check this – I consider it to be a geographical factor.

Lines 71-73: “best predicted by previous years’ case data from the focal county or its neighboring counties”. This finding is very focal, so what are the implications for rabies control? i.e. look where you've found it recently. This seems like an obvious and simplistic interpretation. More insights are needed to justify publication.

Lines 74-76: “Identification of trends in where and when cases are most likely to occur can help with a better understanding of the factors driving endemic disease dynamics in wildlife, and can help focus education, surveillance, and management efforts on higher risk areas and seasons.“ Based on this statement, knowing that raccoon rabies occurs where and when it has occurred before, and is dependent on the amount of sampling undertaken, the authors need to explain better how their current study meets this goal.

Line 93: “the distribution of cases is far from uniform”. This statement appears to be based on highly biased surveillance data; need to consider this before creating hypotheses for a phenomenon that might not exist.

Lines 96-99: Unfortunately, it is very difficult (impossible?) to test hypotheses of disease dynamics with surveillance data, because surveillance would be expected to be driven largely by disease dynamics in the first place. Rabies is a classic example of this: when it is observed as a clinical disease sampling occurs, when it is not there is little sampling done. This is the issue with using prevalence (the denominator problem) and persistence (haphazard surveillance) as outcomes.

The Introduction section is a bit long, and unfocused e.g. delving into vectors of disease (such as mosquito species), mange, rodent-borne pathogens. It is all interesting, but not necessarily relevant to the study conducted.

Discussion

Lines 407-410: “We find that county-level raccoon rabies dynamics are strongly driven by temporal … and spatial … context, as well as testing intensity … . As above, the importance of these results is more to do with surveillance than rabies epidemiology.

Lines 410-413: latitude and urbanness seem to play a relatively minor role once time, space and sampling taken into account. This conclusion does not seem to come through clearly in the manuscript.

A weakness of the analysis is the lack of a population at-risk. I’d like to see a sensitivity analysis of this, do results changed if population distribution maps change?

Lines 431-435: “However, in the north, more raccoons were submitted for rabies surveillance in the summer compared to the winter; therefore, inferring case trends from looking only at the number of positive samples through time could be misleading. In the north, more raccoons might be submitted in the summer than in the winter because humans are more likely to be outside in the summer.” Again, surveillance implications versus rabes epidemiology insights.

A weakness of the analysis - population at-risk. Would like to see a sensitivity analysis of this, do results changed if distribution maps change?

Lines 445-447: “The surprise here was the notable drop in raccoon rabies presence in the summer in the far north, despite increased sampling”. I agree, surprising! Implications for surveillance.

Lines 452-456: “In our analysis, we aggregated across years, possibly obscuring an effect of the time-since-introduction, which varies across the spatial extent of our data. In particular, one might expect that the relatively recent introduction of rabies to the northern states might be consequential to disease dynamics. We do not see any evidence for this in our data.” But weren’t the areas where rabies had been recently introduced excluded from data analysis? So, I find this statement to be confusing.

Lines 469-470: “What was contrary to our expectations was the highest raccoon rabies prevalence rates in the counties with the lowest human population density (rural areas), especially in the south.” This could well be the “denominator effect”. The analysis of prevalence with the data available (even the use of the term) is very problematic.

Lines 482-483:

Our counterintuitive urbanness results might have to do with challenging sampling issues … counties with more people had more raccoons submitted for testing. Many of findings might be explained by sampling bias.

Lines 496-498: “i.e., with few tests, observed prevalence options can take on only a few discrete values so when only one test is submitted in a given month, a positive test carries more weight than when there were many tests submitted.” Exactly! Hence my suggestion not to include prevalence in your analysis.

What about the lack of independence? i.e. once one rabid case identified there is more sampling?

Lines 517-519: It is still not entirely clear to me whether the null hypotheses have been rejected or not. “ultimately we did find a trend with latitude” (but not urbanness?) feels to me like more is being read into the results than is appropriate. If we only focus on presence, then rabies presence is associated with time, place and sampling.

Figure 2

Grey shading is difficult to visualise. Essentially there seems to be no variation in the estimates, presumably because of the large sample size.

Figure 3

‘presence’ the only outcome with apparent seasonality, but it is also highly variable.

Figure 5

Extremely small R2 values, what does this mean? Is it useful to show?

PLOS authors have the option to publish the peer review history of their article (what does this mean? ). If published, this will include your full peer review and any attached files.

**Do you want your identity to be public for this peer review?** For information about this choice, including consent withdrawal, please see our Privacy Policy .

Reviewer #1: No

Reviewer #2: No

**Figure resubmission:**
---

## [Editor Report · Decision Letter 1]

17 Sep 2025

Dear Michalska-Smith,

We are pleased to inform you that your manuscript 'Landscape-scale analysis of raccoon rabies surveillance reveals different drivers of disease dynamics across latitude' has been provisionally accepted for publication in PLOS Neglected Tropical Diseases.

Best regards,

Simon Rayner

Academic Editor

David Safronetz

Section Editor

Shaden Kamhawi

co-Editor-in-Chief

Paul Brindley

co-Editor-in-Chief

The original reviewer who requested a major revision declined to reviewed the new manuscript. However, from reading the authors' detailed response to each of the reviewer's concerns, i believe they have more than adequately addressed all points, and i don't feel it would be appropriate to send out to another reviewer to solicit a further review. This was an interesting and well executed study that i enjoyed reading, thank you

---

## [Editor Report · Acceptance letter]

Dear Michalska-Smith,

We are delighted to inform you that your manuscript, "Landscape-scale analysis of raccoon rabies surveillance reveals different drivers of disease dynamics across latitude," has been formally accepted for publication in PLOS Neglected Tropical Diseases.

Best regards,

Shaden Kamhawi

co-Editor-in-Chief

Paul Brindley

co-Editor-in-Chief
